# The Growth of Ga_2_O_3_ Nanowires on Silicon for Ultraviolet Photodetector

**DOI:** 10.3390/s19235301

**Published:** 2019-12-02

**Authors:** Badriyah Alhalaili, Ruxandra Vidu, M. Saif Islam

**Affiliations:** 1Nanotechnology and Advanced Materials Program, Kuwait Institute for Scientific Research, Safat 13109, Kuwait; eng.kisr@hotmail.com; 2Electrical and Computer Engineering, University of California at Davis, Davis, CA 95616, USA; sislam@ucdavis.edu; 3The Faculty of Materials Science and Engineering, University of Politehnica of Bucharest, 060042 Bucharest, Romania

**Keywords:** β-Ga_2_O_3_, nanowires, oxidation, silver catalyst, electrical conductivity, photodetector

## Abstract

We investigated the effect of silver catalysts to enhance the growth of Ga_2_O_3_ nanowires. The growth of Ga_2_O_3_ nanowires on a P^+^-Si (100) substrate was demonstrated by using a thermal oxidation technique at high temperatures (~1000 °C) in the presence of a thin silver film that serves as a catalyst layer. We present the results of morphological, compositional, and electrical characterization of the Ga_2_O_3_ nanowires, including the measurements on photoconductance and transient time. Our results show that highly oriented, dense and long Ga_2_O_3_ nanowires can be grown directly on the surface of silicon. The Ga_2_O_3_ nanowires, with their inherent n-type characteristics formed a pn heterojunction when grown on silicon. The heterojunction showed rectifying characteristics and excellent UV photoresponse.

## 1. Introduction

The development of wide band gap semiconductor technology has received considerable attention as basic materials that facilitate various ultraviolet (UV) applications in nanoscale electronics and optoelectronics [1] such as engine control, solar UV monitoring, astronomy, communications, or the detection of missiles. Recently, UV photodetectors (PDs) have received special attention, because the civil, military, environmental, and industrial markets need to improve UV instrumentation that operates at extremely harsh environments. Therefore, numerous studies have been proposed to fabricate UV photodetectors with specialized features to operate and survive in the UV region of the spectrum.

Semiconductor nanowires exhibit different and often improved material properties [2,3] compared to bulk or thin-film semiconductors. In recent years, gallium oxide (Ga_2_O_3_) became one of the most important materials that can operate in harsh conditions. With a band-gap of 4.8 eV, a high melting point of 1900 °C, excellent electrical conductivity, high figure of merit for high-frequency applications, and photoluminescence [4,5], it is an ideal candidate for visible-blind UV-light sensors, particularly for power electronics, solar-blind UV detectors, and devices for harsh environments [6,7]. New processes have been investigated to synthesize Ga_2_O_3_ nanowires (NWs) through a bottom-up approach, which include thermal oxidation [8,9], vapor-liquid-solid mechanism [10], pulsed laser deposition [11], sputtering [12], thermal evaporation [13,14,15], molecular beam epitaxy [16], laser ablation [17], arc-discharge [18], carbothermal reduction [19], microwave plasma [20], metalorganic chemical vapor deposition [21], and the hydrothermal method [22,23].

Due to the surface area, small nanowire diameter and high nanowire photoconductivity, high responsivity can be achieved in UV photodetectors. Additionally, one of the beneficial parameters of nanowires is their ability to enhance light absorption and confinement to increase photosensitivity [24]. The superiority of the growth of Ga_2_O_3_ nanowires, compared to thin film is the ability to increase the sensitivity in detection due to the higher surface-to-volume ratio, leading to more available surface states at the interface, and thus, exceptional interaction with analytes or physical states [25]. Although various reports have been obtained to grow Ga_2_O_3_ thin films on Si [26,27], there have been few reports on the growth of nanowires onto a silicon (Si) substrate [28], which will pave the way for future sensing devices and circuit technology integrations. The sensors obtained using this innovative approach will lead to new trends in design, control, and applications of real-time intelligent sensor system control by advanced intelligent control methods and techniques. The effect of Ag thin film as a catalyst to enhance the growth of Ga_2_O_3_ nanowire and crystalline thin film on quartz has been reported [29], but it has not been explored on the silicon surface. We also wanted to observe the contribution of silicon atoms in enhancing the conductivity of Ga_2_O_3_ nanowires via diffusion-enabled incorporation into the nanowires during the growth process.

In our previous work, the surface of the quartz was coated with a 5 nm Ag catalyst using a shadow mask intentionally to examine the effect of Ag nanoparticles (NPs)distribution. In this work, the entire silicon surface was coated with a 5 nm catalyst to enhance the growth of highly oriented nanowires that has not been shown before. Compared to other reported works [28], the length of the nanowires was much higher and highly oriented when Ag catalyst was used rather than the Au catalyst, where the nanowires were randomly oriented.

In this work, we proposed the growth of β-Ga_2_O_3_ nanowires on P^+^-silicon substrate by thermal oxidation at 950 °C using an Ag catalyst. We studied the sensitivity of β-Ga_2_O_3_ nanowires for UV detection.

## 2. Materials and Methods

The UV photodetector was fabricated on (100) P^+^-Si substrate doped with phosphorus. The substrate was 500 μm thick and had a resistivity between 0.001 and 0.005 Ω-cm. Before each experiment, the silicon substrate was cleaned for 5 min in acetone and then, in methanol for 5 min in an ultrasonic bath. Following the cleaning procedure, the wafer was rinsed with deionized water for 5 min. To obtain Ga_2_O_3_, 0.2 g of gallium [(Ga) (purity 99.999%)] was dripped onto cleaned quartz crucible. Silver was used as a catalyst to enhance the growth of gallium oxide NWs. An ultrathin layer of 5 nm Ag was sputtered on silicon. The silicon wafer was positioned with the Ag-coated surface to face the crucible quartz containing Ga. The distance between the substrate and the gallium pool was 10 mm. Then, the substrate was loaded into a quartz crucible which was placed into an OTF-1200X-50-SL horizontal alumina tube furnace made by MTI Corporation (Richmond, CA, USA). The oxidation was performed at 950 °C for 1 h in a 20 sccm nitrogen atmosphere.

Figure 1 illustrates the setup of the UV photodetector fabrication process. As the system cools down to room temperature, the samples were removed from the furnace, cleaved, and characterized by scanning electron microcopy (SEM), X-ray photoelectron spectroscopy (XPS), and high-resolution transmission electron microscopy (HRTEM), equipped with energy dispersive X-ray spectroscopy. The electrical contacts were patterned on top of the nanowires using shadow mask and then 1 nm Cr and 150 nm Au were sputtered using a Lesker sputtering system. Electrical characterization of the system was also carried out to assess the performance of the UV photodetector. For electrical measurements, a custom probe station attached to a Keithly 2400 series SMU instrument was used. For photocurrent measurements, UV illumination was from a Dymax Bluewave 75 UV lamp (280–320 nm) (Dymax Corporation, Torrington, CT, USA). A light intensity of 1.5 W/cm^2^ was used.

## 3. Results and Discussion

### 3.1. Surface Morphology

Ga_2_O_3_ nanowires were grown on P^+^-Si at 950 °C. As shown in Figure 2, the silver catalyst plays a major role in the growth mechanism. Using 5 nm Ag as a catalyst, a homogeneous coating and denser nanowires were achieved due to the low contact angle. A low contact angle reflects the extension of wetting, i.e., the liquid advances on the surface and homogeneously wets the surface. To control the wetting contact angle, deposition or incorporation of elements and molecules onto the surface is a standard procedure. We believe that Ag has the role to improve wettability, which will enhance the homogeneous appearance of Ga_2_O_3_ nuclei that could lead to dense nanowires. The contact angle of Ga on a silver film is 30° [30], and on a silicon substrate, it is 73.9° [31], leading to better wetting of Ga on Ag surface and uniform growth of Ga_2_O_3_ nanowires (Figure 3).

Various research strategies were conducted in the past, mainly to enhance the nanowires’ growth on the target substrate [10,32,33,34]. In contrast, these techniques to grow Ga_2_O_3_ nanowires have shown lateral growth, overlapping nanowires, less dense and weak adhesion to the substrate. None of the previous techniques were able to produce a conformal growth process of Ga_2_O_3_ nanowires on the substrate surface.

The results obtained with the use of 5 nm Ag catalyst showed a remarkable improvement in the lengths and the density of the nanowires, most of them perpendicular to the surface. Even though the lengths of these nanowires were increased, their diameters were decreased. The diameters of the nanowires were in the range of 70–90 nm at the tip and 120–160 nm at the bottom. The average length of these nanowires was in the range of about 30–70 µm.

### 3.2. X-ray Photoelectron Spectroscopy (XPS)

To analyze the elemental composition of Ga_2_O_3_ nanowires, XPS was performed on a PHI 5800 model.

Figure 4 shows XPS spectra of Ga_2_O_3_ nanowires on Si. The XPS spectrum shows the chemical composition of the particles at the surface of β-Ga_2_O_3_ nanowires on Si in the presence of Ag. The binding energies of Ga2p3, O1s, and Ag3d (with two peaks) and Si2p are 1119.1 eV, 532 eV, 369.07 eV and 379.66 eV and 105.18 eV, respectively. The peaks of Ga and O for Ga_2_O_3_ and Ag are in agreement with the handbook of XPS spectra [35,36]. XPS analysis of the β-Ga_2_O_3_ nanowires on Si and the presence of Ag catalyst showed a positive shift due to the effect of the electronegativity difference [37]. In addition, this shift could be attained in Ag3d, as the size of Ag nanoparticles highly decreased [38].

### 3.3. High-Resolution Transmission Electron Microscopy (HRTEM)/Energy-Dispersive Spectroscopy (EDS)

An energy-dispersive spectroscopy (EDS) profile analysis was performed on β-Ga_2_O_3_ nanowires grown on Si (Figure 5). Interestingly, none of the Ag nanoparticles were clearly observed on the surface of the nanowires. However, a very small amount in atomic percentage of Ag was detected by HRTEM equipped with EDS. Because no Ag was observed on the nanowire surface, a very small amount of Ag might be embedded into the Ga_2_O_3_ nanowires. These remaining Ag nanoparticles could be trapped inside the nanowires after all Ag was consumed and evaporated.

Because silicon atoms can interact with silver at a high temperature (i.e., the oxidation temperature of 950 °C) the background impurity of silicon was measured in Ga_2_O_3_ nanowires. At high temperature and a few atomic percentages of Si, the Si-Ag phase diagram [39] shows that Si can interact with Ag. Silicon is one of the major impurities that strongly correlates to n-type conductivity [40]. If silicon were to be incorporated into Ga_2_O_3_ nanowires during oxidation, it could increase the n-type conductivity of nanowires. In addition, since Si has a strong effect on the dissolution of the large Ag NPs [41], there will be more Ag atoms available for diffusion on the Si surface, which could result in a denser growth of nanowires.

### 3.4. Growth Mechanism of β-Ga_2_O_3_ Nanowires

The contribution of a silver catalyst to the growth enhancement of β-Ga_2_O_3_ nanowires on Si showed a growth reaction rate strongly influenced by the oxidation temperature and follows the Arrhenius law [42]. Oxygen diffusivity and solubility are important parameters that distinguish Ag as an effective catalyst for Ga_2_O_3_ nanowire growth.

Diffusion is a result of the kinetic properties of atoms. In this case, diffusion appears to be due to the high capability of Ag to absorb oxygen, and it is greatly influenced by the variation of temperature. Different studies were focused on the oxygen diffusivity (D) in gallium [43] and silver [44]. Table 1 summarizes the major diffusivity coefficient of oxygen into solid Ag, liquid silver, and liquid gallium. The diffusion coefficient of oxygen in silver has a high tendency to absorb oxygen, and hence, boost nanowire growth.

The solubility of oxygen is another factor that has essential perspective to speed up the growth of Ga_2_O_3_ nanowires. The activation energy of oxygen solubility in silver was 0.01192 eV/K at a temperature range of 763–937 °C [45]; however, in gallium, it was 2.38 × 10^−4^ eV/K at a temperature range of 750–1000 °C [48]. Oxygen solubility in silver exhibited a higher solubility than Ga. Further studies are needed to measure the Ag-Ga-O thermodynamics at higher temperatures.

Taking these results into consideration, the growth mechanism of nanowires can be explained as follows. First, at higher temperatures, the liquid gallium can form gallium oxide in the presence of oxygen. Then, the oxide is further reduced by liquid metallic gallium and forms a gas phase of gallium suboxide (Ga_2_O), as shown in Equation (1) [29,49] as follows:Ga_2_O_3(s)_ + 4Ga_(l)_ → 3Ga_2_O_(g)_ ↑(1)

The Ga_2_O gas phase is transported to the cooler regions and decomposes to liquid gallium and Ga_2_O_3_ [50,51], leading to a vapor-liquid-solid (VLS) growth mechanism. At high temperatures (T > 950 °C) denser Ga_2_O_3_ grows as nanowires. It has been shown that the presence of Ga atoms can easily etch the surface of silica substrate around 950 °C, as shown in Equation (2) [52].
SiO_2_+ 4Ga→ 2Ga_2_O ↑+ Si(2)

In addition, the phase diagram of Si-Ag shows that a liquid phase exists in this system at high temperatures (T > 800 °C) at a small percentage of Si [39]. The contribution of small concentrations of silicon can detach and stimulate the melting point of Ag surface atoms [41]. Despite the fact that carrier doping in β-Ga_2_O_3_ is a difficult task, some impurity doping using Sn or Si has been shown to achieve electrical conduction [40,53,54,55]. In this growth mechanism, silicon has been detected by EDS (Figure 5), unintentionally improving the background conductivity of the nanowires. The presence of oxygen atoms segregated on the surface of Ag catalyst will react with Ga. This increases the flux of O atoms and Ga segregation at Ag-Si interface, leading to the formation of an equilibrium mixture of Ag-Si-Ga-O that becomes a solid phase source for Ga_2_O_3_ nucleation (Figure 6).

### 3.5. Electrical Characterization

#### 3.5.1. I-V Characterization

The β-Ga_2_O_3_/P^+^-Si PN heterojunction (Figure 7) was fabricated to determine the electronic properties of β-Ga_2_O_3_ nanowires. The choice of testing the P^+^Si substrate is due to availability of low-cost materials for electronics and to observe how silicon from the substrate can influence the conductivity of Ga_2_O_3_. Impact of silicon doping in Ga_2_O_3_ during the growth processes were reported in references [40,53,54,55] for the cases of thin films and bulk materials and we wanted to investigate if migration of silicon atoms from the substrate can have a similar effect. In addition, the formation of n-Ga_2_O_3_ nanowires on the surface of highly doped silicon substrates has not been reported so far. The results lead to the development of a simple growth technique for large-scale production of a highly sensitive and stable structure. In previous works, the growth of Ga_2_O_3_ was obtained due to the presence of an Au catalyst (instead of Ag) on the surface of the Si_2_/Si template [28].

The current-voltage (I-V) characteristics were measured in dark conditions and under UV illumination at different voltages 10 and 50 V. Photocarriers, which were excited by UV illumination, were from a Dymax Bluewave 75 UV lamp (280–320 nm) (Dymax Corporation, Torrington, CT, USA) (Figure 8). The photoconductivity mechanism of the β-Ga_2_O_3_ NWs is credited to a surface oxygen adsorption and desorption process [56], which is highly influenced by the presence of silver as a catalyst, leading to improve oxygen detection and hence the electrical properties of the β-Ga_2_O_3_ nanowires.

The ratio of photo-to-dark current at 10 V was 3066.11 which is higher than other reported studies [57,58]. The reduction in performance could be attributed to the presence of Ag NPs, which were detected by XPS, although they are difficult to see in the scanning and transmission microscopy (SEM) images. The hot carriers of Ag NPs could increase the self-heating effects [59]. This issue is one of the major challenges that is still under investigation to improve the thermal conductivity of Ga_2_O_3_. It is well known that Ga_2_O_3_ generates self-heating effects that cause degradation of the carriers mobility [60], leading to reduced performance of Ga_2_O_3_ at high voltage.

Even the addition of Ag catalyst could cause a drawback, as it can enhance the sensitivity of the photodetector. The effect of the catalytic Ag nanoparticles can be explained as follows. First, Ag nanoparticles have a significant contribution in improving the conductivity of Ga_2_O_3_ nanowires, leading to better sensing performance. Secondly, Ag nanoparticles have the ability to greatly enhance the adsorption and desorption of O_2_ on their surface due to the highly conductive behavior of Ag metal [61]. Consequently, the number of electrons drawn to O_2_ increases greatly. Third, Ag nanoparticles play the role of electron mediators that allow electrons to migrate from the surface of Ga_2_O_3_ nanowires to the O_2_ through the defect states of Ga_2_O_3_. As a result, the bulk defects of Ga_2_O_3_ may act as a secondary factor in the sensing mechanism in addition to the surface defects [4]. Consequently, Ag NPs significantly reduce the density of electrons of Ga_2_O_3_ and improve electrical conductivity, leading to better selectivity and sensitivity.

#### 3.5.2. Transient Time

The transient response of the photodetector was measured by turning on and off a UV light source with wavelength range from 280 to 450 nm (Figure 9). Under UV illumination, the oxygen adsorption and desorption processes are attained to improve the photoconductivity response by increasing the carrier mobility. In contrast, when UV illumination is switched off, the excess electrons and holes recombine rapidly. Ga_2_O_3_ on silicon with Ag catalyst showed a rapid transient response due to the enhanced carrier transport. The rise was 0.8 s and fall time was 1.5 s. Due to the enhanced carrier transport process, fast rise and decay of the photocurrent were obtained.

#### 3.5.3. Detection Mechanism

In dark current measurements, Ag NPs cause a localized Schottky junction and deplete the carriers at the interface of β-Ga_2_O_3_ nanowires. Therefore, there is a large depletion width at the interface between Ag NPs and β-Ga_2_O_3_ nanowires, leading to a decrease in the dark current of the UV photodetector.

UV detection mechanism is determined based on the contribution of two different parts, namely, Ag nanoparticles catalyst and P^+^-silicon. Under UV illumination, when the photon energy is larger than the bandgap of Ga_2_O_3_, carriers (electron-hole pairs) are generated [*hv* → e^−^ + *h*^+^]. These enhanced photo-generated carriers by the large electric field increase the carrier density in β-Ga_2_O_3_ nanowires and improve the photocurrent response. The energy band diagrams of the AgNPs/ β-Ga_2_O_3_ /p-Si p-n junction is shown in Figure 10a. The band offsets values are estimated using the electron affinity 4.05 eV [62], 4.00 eV [63], and band gap 1.12 eV, 4.9 eV for p-Si, and β-Ga_2_O_3_, respectively. The work function (φ_Ga2O3_) and electron affinity (χ_Ga2O3_) of β-Ga_2_O_3_ are 4.11 eV and 4.00 eV [63], respectively. This is lower than the work function of Ag (4.26 eV), leading to the formation of a Schottky barrier which prevents the electrons transport from Ag NPs side to Ga_2_O_3_. In addition, Ag NPs on the surface of Ga_2_O_3_ is highly influenced with UV light below 320 nm due to the interband transitions, exciting the transition of highly energetic hot electrons from the 4d and 5-sp bands [64,65,66]. These hot electrons surmount the small height of Schottky barrier and lead to local band bending downward on the Ga_2_O_3_ side to enable the electron transfer to the conduction band of Ga_2_O_3_ nanowires.

Regarding the silicon contribution, when the applied voltage is positive on Ga_2_O_3_, the movement of holes can easily be achieved; hence, photocurrent response is increased. However, if the voltage is negative, the holes are constrained and cannot jump the hill to the side of p-Si. Consequently, the presence of more electrons can increase oxygen molecules absorption and ionization [O_2_+ e^−^ → O_2_^−^ [ad]] [67,68]. However, the holes drift to the surface, accumulate, recombine with adsorbed ionized oxygen and form free oxygen molecules from the surface [O_2_^−^ [ad] + h^+^ → O_2_]. The remaining electrons become the majority carriers that contribute to an increase in the photocurrent by generation and recombination until reaching an equilibrium phase.

Nanowires offer a great opportunity to form a higher density of exposed surface states due to the dangling bonds at the surface of nanowires. These trap states of oxygen generated at the surface of Ga_2_O_3_ nanowires have a large impact on device performance [2]. The detector can be easily and fully integrated on a chip with proper metal contacts similar to the graphene-based detectors [69]. Due to the large surface to volume ratio of nanowires and the existence of Ag NPs, the surface of NWs with trapped oxygen becomes highly sensitive.

## 4. Conclusions

Highly oriented, dense, and long β-Ga_2_O_3_ nanowires were grown on P^+^-Si (100) substrate in the presence of a 5 nm thin film of Ag catalyst and oxidation treatment at high temperature (1000 °C). Silver was shown to have a great impact to expedite the growth of Ga_2_O_3_ nanowires and retain their physical and chemical properties. The morphological, compositional, and electrical properties were explored. The growth mechanism of nanowires on the silicon substrate was discussed. During the growth process, Ga_2_O_3_ nanowires are highly influenced by silicon as unintentional impurities that increase the n-type doping. The photoresponse under UV irradiation was excellent. The ratio of photo-to-dark current (I_photo_/I_dark_) was measured to be around 3.07 × 10^3^ at 10 V. The high photosensitivity could be attributed to the higher electron density in Ga_2_O_3_ nanowires with Ag NPs. The carrier transport process was shown to have a fast response. The energy band gap and carrier dynamics at the interfaces were discussed. This synthesis can be optimized for sensing, electronics, and photonic applications.

## Figures and Tables

**Figure 1 sensors-19-05301-f001:**
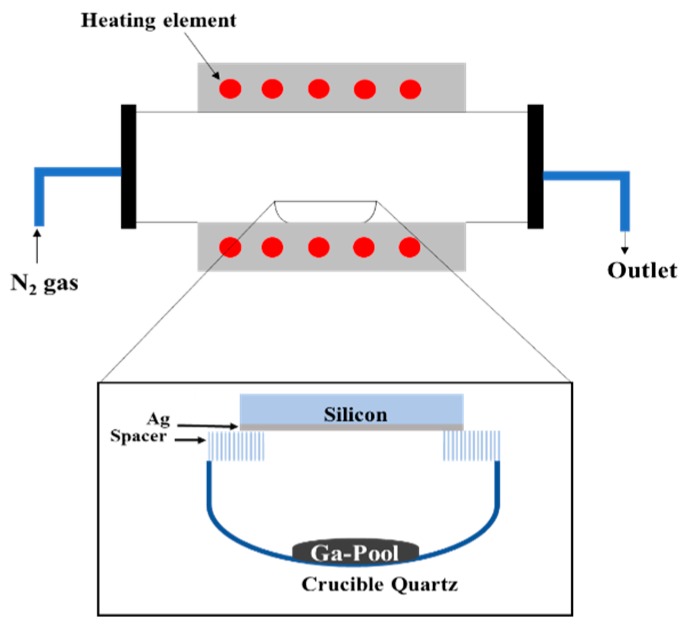
Schematic of the growth process of Ga_2_O_3_ NWs on Si substrate coated with 5 nm thin film of Ag and positioned downward to face liquid Ga pool in a quartz crucible. The distance between Ga pool and silicon substrate is about a ~10 mm gap.

**Figure 2 sensors-19-05301-f002:**
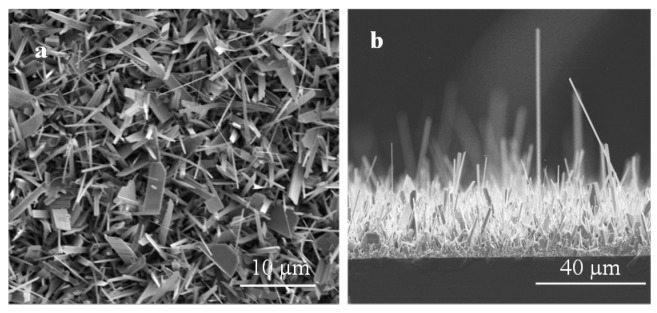
SEM images of Ga_2_O_3_ nanowires growth on Si at 950 °C (**a**) Top view and (**b**) Side view of Ga_2_O_3_ nanowires growth on Si. Denser and longer growth of nanowires were attained.

**Figure 3 sensors-19-05301-f003:**
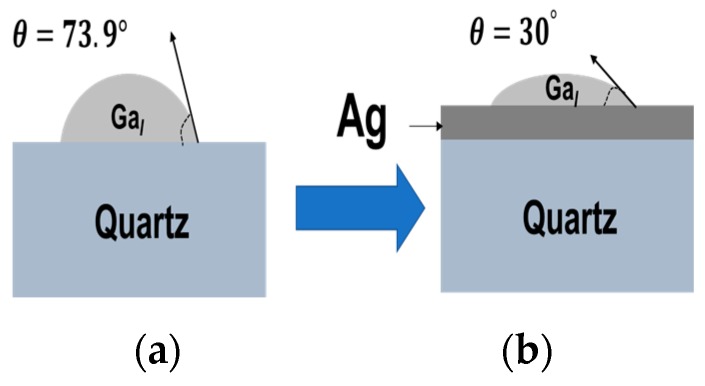
Contact angle of liquid Ga droplet on different surfaces. (**a**) Silicon. (**b**) 5 nm silver thin film. Areas coated with 5 nm Ag show uniform and high-dense growth of Ga_2_O_3_ nanowires.

**Figure 4 sensors-19-05301-f004:**
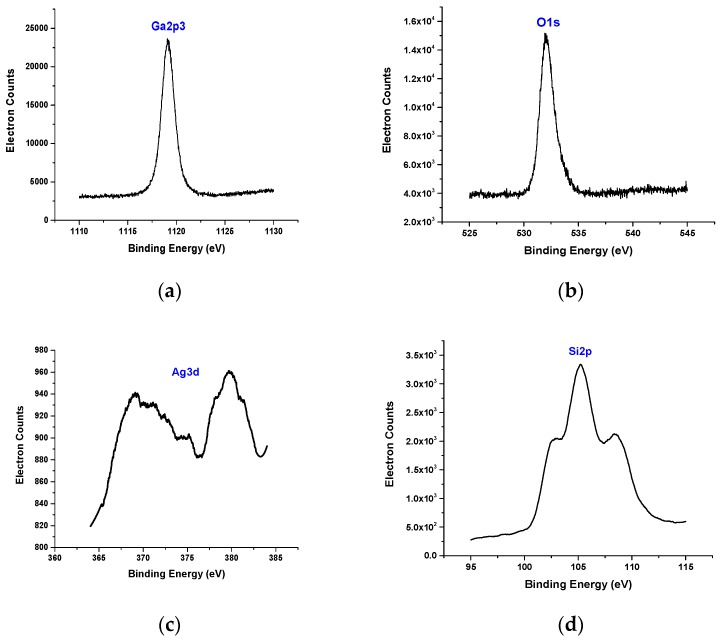
XPS of the β-Ga_2_O_3_ nanowires was obtained at 950 °C and in the presence of an Ag catalyst. Different peaks were detected by XPS. (**a**) Ga. (**b**) O. (**c**) Ag. (**d**) Si. The peaks of Ag and Ga have positive slight shifts due to the difference in electronegativity and work function.

**Figure 5 sensors-19-05301-f005:**
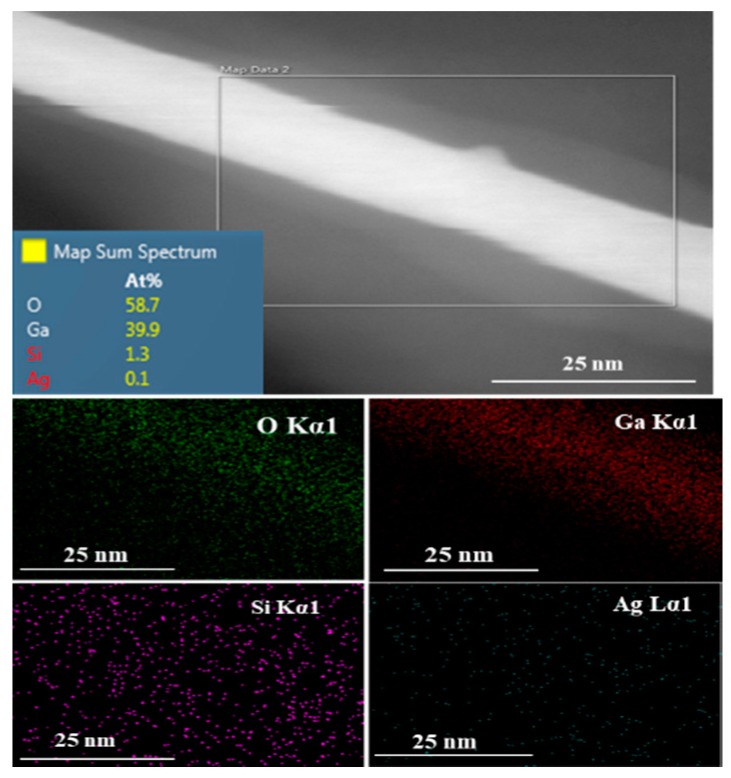
HRTEM image and the corresponding EDS mapping of Ga, O, Si and Ag of Ga_2_O_3_ NWs on P-doped (100) silicon substrate coated with 5 nm Ag.

**Figure 6 sensors-19-05301-f006:**
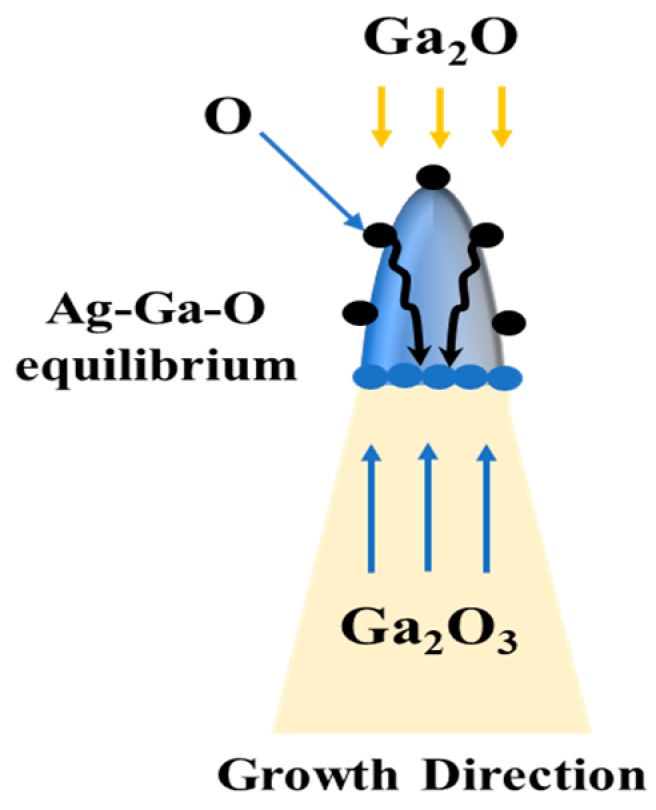
The growth mechanism of Ga_2_O_3_ NW on a silicon substrate coated with 5 nm Ag as a catalyst. The equilibrium liquid mixture of Ag-Ga-O at higher temperature (>900 °C) leads to the enhancement of the growth mechanism and increases the density of Ga_2_O_3_ NWs.

**Figure 7 sensors-19-05301-f007:**
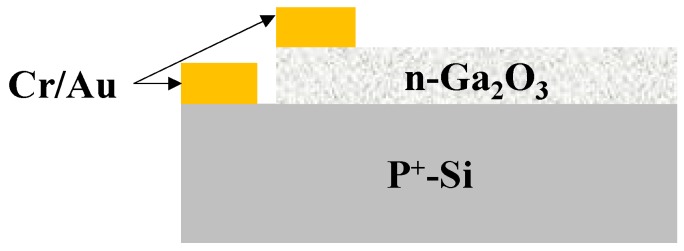
Schematic diagram of Au/β-Ga_2_O_3_/Silicon photoconductor. The distance between the gold probes is 0.8 mm.

**Figure 8 sensors-19-05301-f008:**
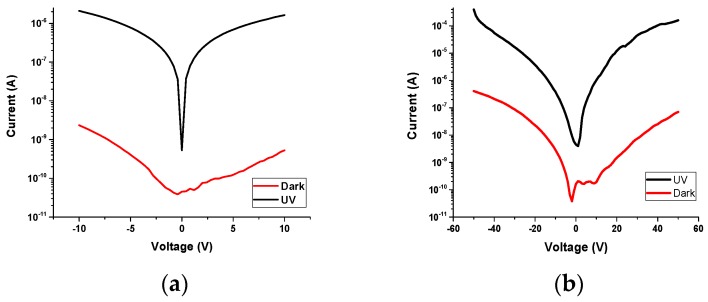
Semi-logarithmic plots of current density of dark and photocurrent characteristics of Ga_2_O_3_ NWs grown on silicon substrate at 950 °C with an Ag catalyst at 10 V, (**a**) 10 V, (**b**) 50 V.

**Figure 9 sensors-19-05301-f009:**
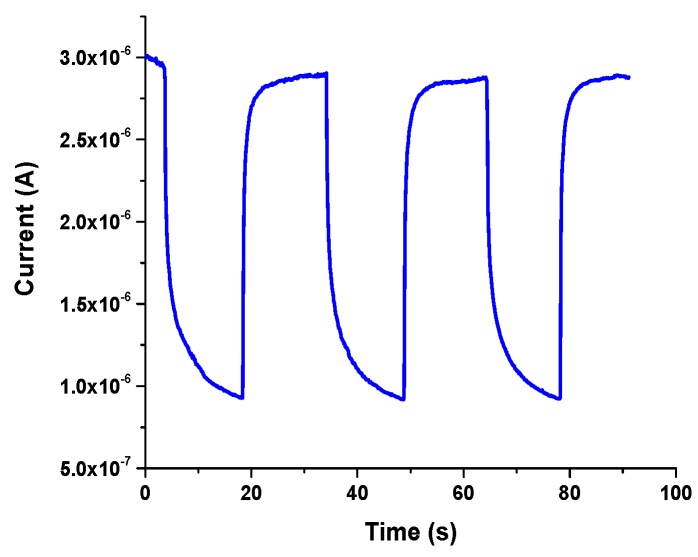
Transient response of the UV photodetector fabricated with Ag catalyst based on Au/β-Ga_2_O_3_/Silicon photojunction at 10 V.

**Figure 10 sensors-19-05301-f010:**
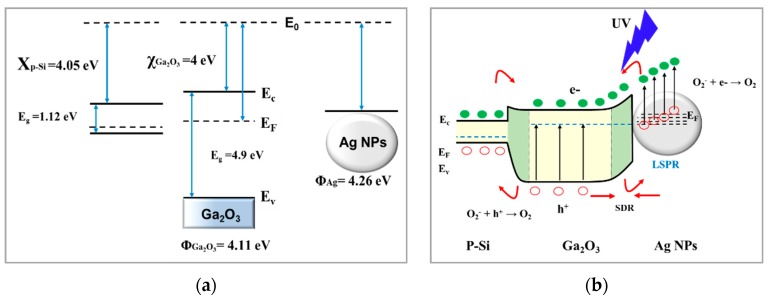
Energy band diagram of Ag NPs and Ga_2_O_3_ NWs pn P^+^-Si. (**a**) at the interface before contact. (**b**) Under UV illumination, the interband transition in Ag NPs enhances the photosensitivity of the UV detection, and more photo-generated holes of Ga_2_O_3_ NWs migrate to the surface by band bending.

**Table 1 sensors-19-05301-t001:** Summary of reported diffusivity coefficient and activation energy of oxygen in silver and gallium.

Metal	Diffusion Coefficient (D)(cm^2^/s)	Activation Energy (E_A_) (eV/K)	T (°C)	Year	Ref.
**Ag*_s_***	1.79 × 10^−3^	0.58	127–977	2016	[44]
	4.90 × 10^−3^	0.56	740–915	1972	[45]
	3.66 × 10^−3^	0.48		1990	[46]
	4.98 × 10^−3^	0.63		1991	[47]
**Ag*_l_***	20.1 × 10^−4^	0.91	980–1130	1971	[9]
	4.9 × 10^−3^	0.12	763–937	1972	[45]
**Ga*_l_***	4.1 × 10^−3^	8.9 × 10^−5^	750–950	1981	[43]
	2.27 × 10^−3^	8.33 × 10^−5^	750–1000	1972	[48]

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
