# Peer review of "The Growth of Ga2O3 Nanowires on Silicon for Ultraviolet Photodetector"

_sensors, 2019, doi:10.3390/s19235301_

Round 1

Reviewer 1 Report

The authors studied the effect of silver catalysts to enhance the growth of Ga2O3 nanowires for developing UV photodetectors. The results are interesting. But the authors should answer the following questions before the manuscript can be accepted.  

In line 185 of Page 6, the authors described the photodetector structure which is based on the metal-oxide-semiconductor (MOS) transistor configuration. The authors should explain where are the gate, drain, and source electrodes in Figure. 7. Besides, the authors should comment on how to tune the gate voltage and obtained the I-V curves in Figure. 8? In Section 3.5.3 of Page 8, the authors claimed that the UV detection mechanism is determined based on the contribution from two parts: Ag nanoparticles catalyst and P+ silicon. But which one dominates the photoresponse? I understand that the authors discuss the detection mechanism in Section 3.5.3 of the manuscript. However, how did the authors measure the energy band diagram? Semiconductor nanomaterial-based photodetectors have attracted great attention. Is it possible to integrate the proposed technique on a chip similar to graphene-based detectors (e.g. Nanoscale, vol. 9, no. 40, pp. 15576–15581, 2017)? The authors should comment on it in the manuscript. The authors should carefully check the language of the manuscript. I found numerous grammatical mistakes and typographical errors in the manuscript. For example, Line 12 “a a thin”, Line 23 “a basic materials”, Line 226 “1.5s”, Line 282 “photoreponse”, Figure 3 caption “on different surface”.

Reviewer 2 Report

The manuscript “The growth of Ga2O3 nanowires on silicon for ultra-violet photodetector” write by Badriyah Alhalaili have investigated the effect of silver catalysts to enhance the growth of Ga2O3 nanowires. The growth of Ga2O3 nanowires is demonstrated by using thermal oxidation technique and shows rectifying characteristics and excellent UV photoresponse. However, the preparation of of Ga2O3 nanowires photodetector already has been reported (Y. L. Wu et al., "Ga2O3 Nanowire Photodetector Prepared on SiO2/Si Template," (in English), Ieee Sensors Journal, vol. 13, no. 6, pp. 2368-2373, Jun 2013.). Also, the use of silver catalysts to enhance the growth of Ga2O3 nanowires by thermal oxidation technique have been reported (Badriyah Alhalaili, etc. Nanomaterials 2019, 9, 1272). Even though they get good morphology and excellent UV photoresponse of their Ga2O3 nanowires, I think this work lack of novelty.

Page 2, line 47, even though there are few reports on the growth of nanowires onto a Si substrate, the superiority of Ga2O3 nanowires compare to Ga2O3 thin films need to be stated. In this work, the authors use silver as catalysts to enhance the growth of Ga2O3 nanowires using thermal oxidation technique. Another very similar work from the same group (Badriyah Alhalaili, etc. Nanomaterials 2019, 9, 1272) also have used Ag as a catalyst to grow Ga2O3 nanowires by thermal oxidation technique, however, the author didn’t refer to this work and did not show enough novelty in this work. In point 2 (Materials and Methods), the author only provides the method for the preparation of Ga2O3 nanowires, however, more details about the materials and detection process are missing. In the first time that “EDS” and “UV” appeared, please give the full name of it. Page 2 Line 81 and Page 3 Line 82, why a low contact angle can make the coating and denser nanowires to be more homogeneous? The format of reference 12 is not correct.

Round 2

Reviewer 2 Report

The authors have given good responds for the questions about what we put forward last time. I think now it is much better than the previous version and this paper can be accepted after the following questions have been solved.

1) The author didn't give a respond to the first paragraph of our review comment, among them, the most important is the novety of this work. So I think that the author should give a state of the novety point by point.

2) As the author state in question 2, the novelty of this work is related to "Ga2O3 nanowires grown on P+ Si substrate and to the investigation of the effect of Ag NPs in enhancing the growth of highly-oriented NWs that has not been shown before", however, Ga2O3 nanowires grown on Si substrate have been reported before, and the author should speak about why they use P+ Si as substrate, not only due to P+ Si substrate do not be used before.

3) It is better that all the exact positions of the changes be also given in the responding letter.
